# Changes in Skeletal Muscle Troponin T and Vitamin D Binding Protein (DBP) Concentrations in the Blood of Male Amateur Athletes Participating in a Marathon and 100 km Adventure Race

**DOI:** 10.3390/ijerph20095692

**Published:** 2023-05-01

**Authors:** Jacek Borkowski, Tadeusz Stefaniak, Piotr Cych

**Affiliations:** 1Department of Physiology and Biochemistry, Wroclaw University of Health and Sport Sciences, 35 J.I. Paderewski Avenue, 51-612 Wroclaw, Poland; 2Department of Immunology, Pathophysiology and Veterinary Preventive Medicine, Wrocław University of Environmental and Life Sciences, C.K. Norwida 31 Str, 50-375 Wrocław, Poland; tadeusz.stefaniak@upwr.edu.pl; 3Department of Sport Didactics, Wroclaw University of Health and Sport Sciences, 35 J.I. Paderewski Avenue, 51-612 Wroclaw, Poland; piotr.cych@awf.wroc.pl

**Keywords:** skeletal troponin T, vitamin D-binding protein, globulin Gc, marathon run, adventure race, muscle injury

## Abstract

This study assessed changes in creatine kinase (CK) activity and skeletal muscle troponin T (sTnT) concentrations in the blood, to estimate the degree of muscle degradation after exercise. In addition, the concentration of vitamin D binding protein (DBP) in the blood was assessed. DBP concentrations were measured in blood as a marker for plasma load by monomeric actin. The study included marathon (MR) participants and 100 km adventure race (AR) participants, who were examined before and after the race. There was a significant (16-fold) increase in CK activity among AR participants, and a significant increase in sTnT concentration―127% in the MR group and 113% in the AR group, while there was a statistically significant decrease in DBP concentration by 14% in the AR group. In addition, it was observed that the initial concentration of DBP in both groups was in a normal range, but was lower than the average population, and the DBP concentration in the AR group was lower than in the MR group. It was concluded that exhausting physical effort such as a marathon or adventure races causes muscle damage with a far stronger influence on sarcoplasm than on filaments. The short-term and slight reduction in the concentration of DBP in blood after such efforts may be due to the appearance of monomeric actin in plasma.

## 1. Introduction

In the past 30 years, marathons and other events that require long-term exercise have become increasingly popular [1,2]. Alongside well-trained athletes, such competitions feature people whose ambition is just to reach the finish line within a stipulated time, or just to reach it at all. The expenditure of such a long and relatively intense physical effort by inadequately trained participants can cause their organisms to suffer severe homeostasis disruption. There is a large number of publications describing the changes of parameters that characterize the organism’s response to such an exhausting physical effort. Some of these parameters, among others, are enzymes/proteins that originate from damaged muscle fibers, e.g., creatine kinase [3,4], lactate dehydrogenase [5,6], fatty acid binding protein [7] and myoglobin [4,8]. Some of the proteins released from damaged muscle fibers come from contractile filaments of these cells, e.g., myosin fragments [9,10], troponin I [11] or actin [12]. There are also several proinflammatory cytokines such as interleukin-6 [13] and acute phase proteins such as C-reactive protein [14] or orosomucoid [15]. Cellular proteins released from damaged muscles into the extracellular space are a problem for the body and should be removed. Particularly troublesome is the presence of an elevated concentration of actin in blood plasma. Actin has the ability to form filaments, which could initiate the formation of blood clots. The human organism has a system of proteins whose function is to prevent the polymerization of actin and its removal from the blood. The main components of this system are gelsolin and the vitamin D binding protein (DBP), also called the Gc globulin [16]. Only a small number of reports describe how the human body tolerates the unusual presence of muscle proteins in the blood [17,18]. The changes of gelsolin concentration as a result of single exercise was demonstrated by Tékus et al. [15] and Ohsawa and Kimura [19]. Our study aimed at highlighting the bodily reaction to skeletal muscle damage caused by exhausting physical exercise. To examine the phenomenon of muscle damage and the bodily reaction to it, several parameters characterizing the degree of damage in muscle tissues have been selected, such as creatine kinase and skeletal muscle troponin T, as well as a protein involved in the removal of monomeric actin from the blood—the vitamin D binding protein.

Troponin T from the skeletal muscle (sTnT) is an example of a miofilamentary protein. Creatine kinase (CK) is an enzymatic protein that originates from sarcoplasm and the vitamin D binding protein (DBP) is a blood plasma protein capable of forming complexes with the monomeric actin [19]. DBP-actin complexes are removed from the blood by Ito cells [20]. To investigate the efficiency of the removal of muscle proteins that entered the blood after exercise, it was necessary to examine the response of an organism subjected to great physical effort. As a stimulus that causes extensive muscle damage, two types of long-term physical activity were chosen—a marathon run and a 100 km distance adventure race.

The aim of the presented study was to assess changes in creatine kinase (CK) activity, skeletal muscle troponin T (sTnT) and vitamin D binding protein (DBP) concentrations in the blood, to estimate the degree of muscle damage in marathon (MR) participants and 100 km adventure race (AR) participants.

## 2. Materials and Methods

### 2.1. Participants

Two groups of participants―each comprising amateur athletes―took part in the study. Both groups, of marathon participants (the Wroclaw Marathon, ambient temperature of 18–23 °C) and the group taking part in the adventure race (the Bergson Winter Challenge), were composed of 9 men. Written consent to participate in the experiment was obtained from each volunteer. The study protocol was approved by the institutional ethics committee (Wroclaw University of Health and Sport Sciences) and was in accordance with the Declaration of Helsinki for Human Research. During the interview, all participants declared a mixed diet with animal protein. Participants in both races are admitted to the competition by a doctor. The characteristics of the marathon group and adventure race group are presented in Table 1.

### 2.2. Procedures

The marathon took place among city streets, the profile of the route was flat and the distance was 42.195 km. The participants of the marathon were asked to drink plenty of fluids, and were encouraged to take advantage of the densely spaced refreshment stations. 

The distance of the adventure race was calculated as approximately 100 km (measured as the optimal path between control points). However, the actual distance traveled by the participants of the adventure race was often greater; participants often erred, and as a result had to traverse at least 140 km in variable winter conditions. The race consisted of mountain bike cycling, running along paths and roads and orienteering, abseiling and even navigating through an underground maze―all in very hilly terrain between 250 and 1400 m above sea level. The route profile is shown in Figure 1. Just as for the marathon, the participants of the adventure race had the possibility to take in fluids and eat during the race.

### 2.3. Measures

The CK activity, sTnT concentration and DBP (Gc globulin) concentration of all participants were monitored. Venous blood samples (drawn from the arm) from the marathon runners (MR) and adventure race participants (AR) were taken before the run (sample A) and immediately after the run (sample B). Blood samples were collected in Sarstedt tubes with serous granules (Stamar, Poland) and then allowed to clot at room temperature for 30 min. The samples were then centrifuged for 10 min at 3000 rpm (Eppendorf Centrifuge 5810, Hamburg, Germany). Serum was extracted and then stored in Eppendorf tubes at −30 °C. Samples remained frozen until assays were made. Hematocrit (Ht) levels among the MR and AR group was determined by the centrifugation of blood in the capillaries.

The method of determining DBP levels was in accordance with that described by Borkowski et al. [21]. Briefly, an ELISA sandwich was applied using a self-obtained polyclonal, a monovalent goat IgG anti-human Gc antibody. An affinity purified antibody was used for coating wells and the biotinylated antibody to detect bounded DBP levels. The immune complex was detected by ExtrAvidin-HRP conjugate. All the investigated samples and the standard were examined in triplicate. The average of three measurements was taken as the assay result. The same was conducted for the standard curve.

ELISA evaluations of the skeletal muscle troponin T (or its fragments) in the serum of the analyzed participants were performed in 96-well microtiter plates (MaxiSorp, Nunc). Monoclonal antibodies were first isolated from ascites (JLT12, Sigma-Aldrich) using affinity chromatography on the Troponin-Sepharose 4B column. JLT preparation of ascites was loaded onto the column equilibrated with a 0.1 M phosphate buffer with a pH of 7.4. The column was washed with the same buffer containing 0.5 M NaCl and then with water. Specific IgG anti troponin T from human skeletal muscle was eluted from the gel using 0.1 M Glycine-HCL buffer with a pH of 2.2. The pH of the obtained preparation was adjusted to about 7.0. The troponin complex used for the preparation of the column was purified from the rabbit muscle method described by Ebashi at al. [22]. The purified antibody (A_280_ = 0.05) was then diluted tenfold (experimentally established in three independent experiments) and 100 μL of the diluted IgG in 0.05 M carbonate buffer (pH 9.6) was coated onto a plate and incubated for 6 h at room temperature on a shaker set at 10 rpm and then blocked by 0.1% casein solution in the same buffer. To each, after being coated, blocked and washed well, a solution was added which always contained the same amount of rabbit biotinylated skeletal muscle troponin complex labeled using the N-hydroxysuccinimide biotin ester and different amounts of the human skeletal muscle troponin T product of Scripps Laboratories Inc. USA (standard curve) or serum tenfold diluted in phosphate-buffered saline (PBS) with 0.05% Tween 20 and 0.1% casein. There was a competition between biotinylated rabbit troponin and non-biotinyladed human troponin T. After incubation, the plate was washed four times in PBS with 0.05% Tween 20 (PBST), and 100 μL of ExtrAvidin-HRP conjugate (Sigma–Aldrich) diluted 1000x was added to the plate and incubated for one hour at room temperature. The plate was then washed in PBST and o-phenylenediamine (0.4 mg/mL) and 0.3% H_2_O_2_ (*v*/*v*) dissolved in 0.1 M citrate buffer (pH 5.0) were added. The enzyme reaction was stopped after fifteen minutes by adding 100 μL of 1 M H_2_SO_4_ and the absorbance was read at 490 nm (as the primary wavelength) using a Bio-Tek 340 EL spectrophotometer together with KC3 software to calculate the obtained data (both from Bio-Tek Instruments; Winooski, VT, USA). Serum sTnT concentrations were calculated by interpolation from a six-point logarithmic standard curve. All the investigated samples and the standard were examined in triplicate. The absorbance was inversely proportional to the concentration of the standard. The standard curve was linear from 4 to 200 ng/mL. Creatine kinase activity was determined by a kit produced by Biosystems; catalog number—11,791 (Spain). Cardiac troponin I levels were determined in the plasma of adventure race participants by the ELISA BioCheck Company, Inc USA. Cardiac TnI I was determined according to the kit manufacturer’s instructions. Measurements were made to ensure that such a strenuous physical effort as the 100 km adventure race did not damage the AR participants’ heart muscle.

### 2.4. Statistical Analysis

Statistica 13.1 software (StatSoft Inc. Tulsa, OK, USA) was used for statistical calculations. The Kolmogorov–Smirnov test was used to determine the similarity of the distribution of analyzed parameters for normal distribution. The analysis of variance with repeated measurements and the post-hoc Scheffe test was used to determine whether there are differences of the statistically significant parameters we evaluated between MR and AR groups and between studies performed before and after exercise. The statistical analyses used adopted *p* < 0.05 as statistically significant. F is the quotient of the between-group variance and within-group variance. η^2^ is a measure of effect size for use in the analysis of variance. *P*—probability value.

An additional statistical analysis was performed on the data obtained 24 and 48 h after the marathon run. This analysis was performed using the Scheffe test following the analysis of variance with repeated measures (F = 13.6; *p* = 0.000; η^2^ = 0.63).

## 3. Results

The concentrations of cardiac troponin I measured in the blood of AR participants in both pre-race and post-race plasma samples were in the range of normal values.

Changes of hematocrit in both groups were small and not statistically significant, ranging from 43.3 ± 1.66% (before) to 43.7 ± 2.45% (after) in the MR group and 43.1 ± 2.32% (before) to 43.3 ± 3.57% (after) in the AR group.

The results of CK activity (in International Units per liter [U·L^−1^]) and sTnT and DBP concentrations are shown in Figure 2, Figure 3 and Figure 4, respectively. 

It was demonstrated that all variables were characterized by a distribution similar to a normal distribution. 

Statistically significant effects have been demonstrated for CK activity: repeated measurements (F = 59.04; *p* = 0.000; η^2^ = 0.787), groups (F = 19.19; *p* = 0.000; η^2^ = 0.545) and mixed effect (F = 22.11; *p* = 0.000; η^2^ = 0.580). Using the post hoc test, statistically significant differences were found between the sample A and sample B in the group AR. A difference was also demonstrated between the group of AR and the MR group in the analysis of samples B (Figure 2). 

Statistically significant effects have been demonstrated for sTnT concentrations: repeated measurements (F = 31.49; *p* = 0.000; η^2^ = 0.663) and groups (F = 16.69; *p* = 0.001; η^2^ = 0.511). Using the post hoc test, statistically significant differences were found between the sample A and sample B in the MR group and between the sample A and the sample B in the group AR. A difference was also demonstrated between the AR group and the MR group in the analysis of samples A and the difference between the group of AR and the group of MR in the analysis of samples B (Figure 3).

Statistically significant effects have been demonstrated for DBP concentrations: repeated measurements (F = 19.04; *p* = 0.000; η^2^ = 0.543) and groups (F = 12.68; *p* = 0.003; η^2^ = 0.442). Using the post hoc test, statistically significant differences were found between the sample A and sample B in the AR group. A difference was also demonstrated between the AR group and the MR group in the analysis of samples A and the difference between the group of AR and the group of MR in the analysis of samples B (Figure 4).

## 4. Discussion

It has long been known that great physical effort, such as that required when completing a marathon, seriously affects the body of an amateur athlete [23]. For instance, there are many reports describing an increase in enzyme activity, primarily of creatine kinase, including CK MB derived from slow twitching fibers [24], as well as other enzymes such as lactate dehydrogenase or aldolase. Although the CK plasma activity level has been recognized over the years, as the marker of muscle damage is still used as one of most popular parameters of physical exercise diagnostics [25]. Additionally, it has been demonstrated that an increase in parameters indicative of inflammation, such as the C-reactive protein [14,26] or IL-6 [13], often results from such activity. It has been observed that the increase in CK in plasma is preceded by an increase in the activity of such genes as cyclooxygenase, interleukins 6 and 8 and others [27]. Changes in CK activity measured immediately after completing the marathon run were not statistically significant. The reason for this phenomenon is a too short time between the stimulus (exercise) and its effect in the form of an increase in the amount of active enzyme molecules in the blood. Samples obtained in the next few days demonstrated a multiple and statistically significant increase in CK activity (CK 24 h after a marathon run—1294.8 ± 539.6; CK 48 h after a marathon run—901.7 ± 641.8). The increase in CK activity measured immediately after exercise in the AR group was statistically significant. In this case, the physical effort was long enough to observe a reaction in the form of a multiple increase in CK activity. See Appendix A in Appendix A.

Changes in the plasma concentrations of skeletal TnT in the participants of both groups were not large. Comparing these changes to an increase in the concentration of troponin I from skeletal muscle (sTnI) described by Avela et al. [28], it should be noted that the increase in sTnT is much smaller—29% in the marathon group and ~13% in the AR group. Avela et al. [28] describe a brief, over tenfold increase in the skeletal muscle troponin I concentration after participation in a marathon. Similar increases of sTnI in plasma are described by Rama et al. [29] as a result of triathlon participation. Sorichter et al. [12] describe a dependence of the increase in skeletal TnI concentration in plasma, based on the type of physical effort made. Thus, running on a treadmill with the intensity of 70% VO_2_max caused a slight increase in this parameter, while much bigger increases were noted after a downhill run, with no changes noted after concentric exercise. The sTnT concentration changes described in the present study are much less marked, despite the fact that the determination method used in this study does not distinguish between whole molecules of troponin T and its fragments containing the appropriate epitope. Troponin T, together with troponin I and troponin C, forms a complex associated with actin filaments in skeletal muscle. Thus, it can be expected that post-exercise changes of troponin T concentrations in blood will be similar to changes in troponin I concentrations obtained by Avela and Rama. The observed moderate increase in the sTnT concentration may be due to various reasons. sTnT is a protein with a higher molecular weight than TnI, which hinders its release from muscle cells. TnT has a strong affinity for tropomyosin. Tropomyosin is a protein that has twice the molecular weight and forms dimers. Troponin T can also form complexes in plasma with tropomyosin derived from damaged erythrocytes [30].

The vitamin D binding protein (DBP) has not yet been described as a parameter associated with exercise, although there are a lot of articles describing the protein as a parameter of clinical significance in many review articles [31]. Changes in the concentration of DBP in the blood, even after very serious disorders of homeostasis, are usually limited. A few to a dozen percentage points’ decrease in this parameter was observed as a result of shock, trauma and surgery. An increase above the pre-surgery level is often observed during the following days [32]. This could indicate the existence of reserves in the system of actin removal from the blood and confirm the rapid turnover of this protein [33,34]. A decrease in DBP concentration in the plasma indicates that some of the molecules of this protein forms complexes with actin molecules and are removed from the bloodstream by the Ito cells present in the liver [20]. Similarly, the decrease in gelsolin concentration as a result of exercise was described by Yu et al. [35]. In this study, the slight decrease in the concentration of gelsolin—a protein also involved in the removal of actin from blood—was visible only in the case of untrained participants. The results described by Tékus et al. [15] are similar. Moreover, the decrease in DBP concentration after exercise in the present study was small and statistically significant in the AR group. The concentration of DBP at a rest condition in both groups—327 mg/L in the marathon group and 275 mg/L in adventure race participants—was lower than the average DBP concentration in humans described by Kawakami et al. [33] (300 to 600 mg/L) or by Gressner et al. [20] (180 to 784 mg/L).

This lower level of vitamin D binding protein can be explained by the fact that people who regularly prepare for such a physical effort may possess elevated activities of the mechanism of actin removal by DBP. The lower concentration of DBP during competitors’ training periods was observed by Martín-Sánchez et al. [36]. This phenomenon may support the thesis of the increased utilization of DBP but it also may be caused by genetic differences.

An increase in CK activity, a slight increase in the concentration of skeletal muscle TnT and a moderate decrease in the DBP concentration form a picture of the changes taking place in the muscle tissue and blood under the influence of exercise that is low in intensity but very large in volume. The multiple increase in CK activity suggests that exercise associated with adventure races causes significant injury of the cell membrane of the myocyte and is very strong [37]. Limited and short-term increases in sTnT concentration may suggest mild damage of the filaments’ structure. Moreover, a small and brief decrease in the concentration of the vitamin D binding protein (DBP) proves that physical effort does not cause a very large outflow of muscle actin. The second factor normalizing DBP concentration in the plasma may be the increased synthesis of this protein in the liver. Systematic sports training or systematic participation in a competition of this type can accelerate the turnover of DBP, even in the organisms of people who are not professional athletes.

## 5. Conclusions

It has been demonstrated in this study that the muscle damage caused by exhausting exercise is responsible for the limited and short-lived decrease in the vitamin D binding protein concentration and limited and short-lived increase in the skeletal muscle troponin T concentration.

The high concentration of DBP in blood and its rapid turnover is one of the mechanisms supporting homeostasis maintenance in cases of individuals performing exhaustive physical efforts.

## Figures and Tables

**Figure 1 ijerph-20-05692-f001:**
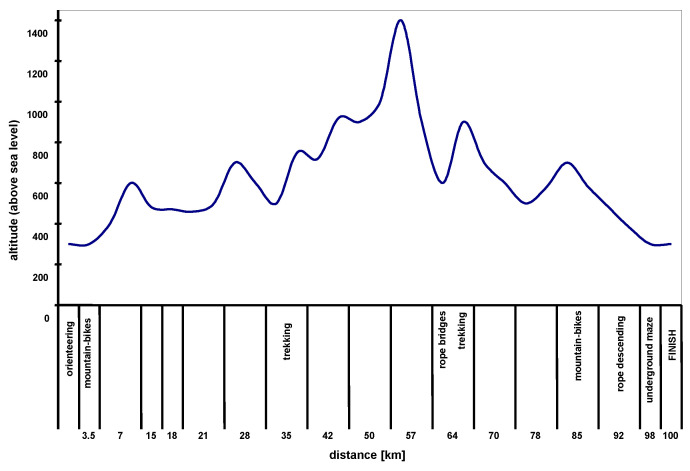
The profile of the adventure race route.

**Figure 2 ijerph-20-05692-f002:**
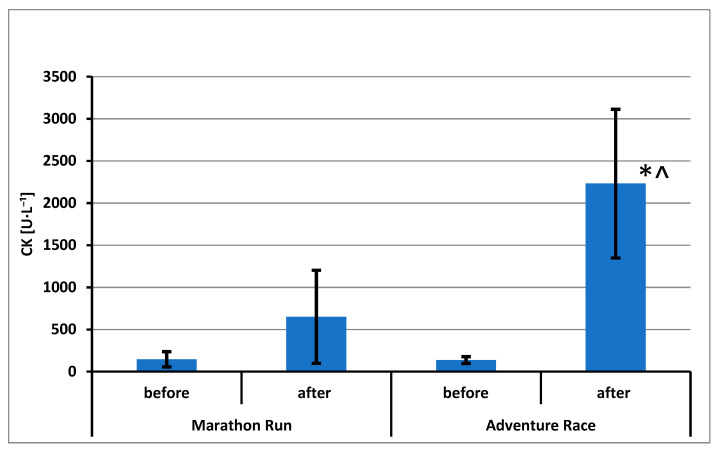
Creatine kinase activity (mean ± standard deviation) in blood of amateur competitors. *—*p* < 0.05 A vs. B; ^—*p* < 0.05 MR vs. AR.

**Figure 3 ijerph-20-05692-f003:**
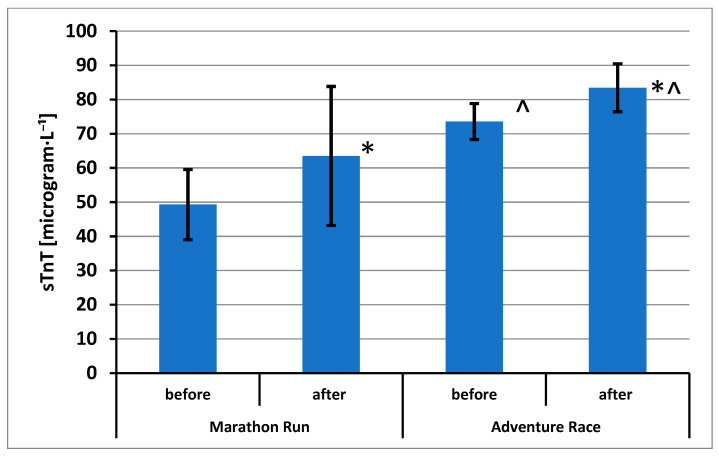
Skeletal muscle troponin T concentration (mean ± standard deviation) in blood of amateur competitors. *—*p* < 0.05 A vs. B; ^—*p* < 0.05 MR vs. AR.

**Figure 4 ijerph-20-05692-f004:**
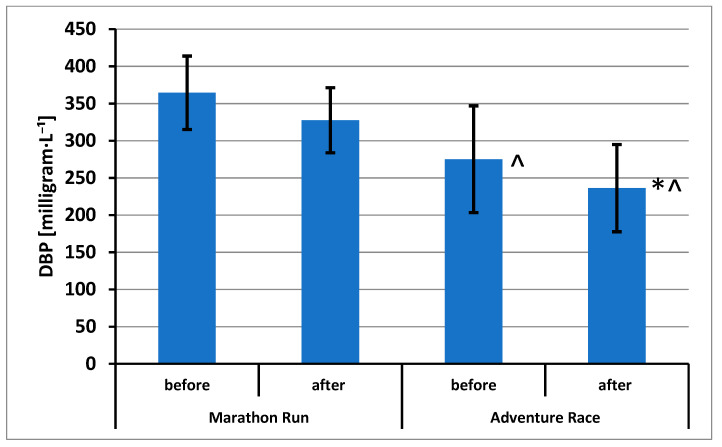
Vitamin D binding protein concentration (mean ± standard deviation) in blood of amateur competitors. *—*p* < 0.05 A vs. B; ^—*p* < 0.05 MR vs. AR.

**Table 1 ijerph-20-05692-t001:** Characteristics of marathon run (MR) and adventure race (AR) participants.

Group	Age [Years]	Body Height [cm]	Body Mass [kg]	Race Result [h]
Marathon Run (MR)	31.7 ± 6.3	180.6 ± 4.6	70.8 ± 2.6	3.5 ± 0.3
Adventure Race (AR)	31.2 ± 6.4	180.4 ± 4.2	71.8 ± 3.7	30.6 ± 3.4

Data are presented as mean ± standard deviation.

## Data Availability

The data presented in this study are available on request from the corresponding author.

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
