# Peer review of "Changes in Skeletal Muscle Troponin T and Vitamin D Binding Protein (DBP) Concentrations in the Blood of Male Amateur Athletes Participating in a Marathon and 100 km Adventure Race"

_ijerph, 2023, doi:10.3390/ijerph20095692_

Round 1
Reviewer 1 Report
Borkowski et al assessed changes in creatine kinase (CK) activity, 66 skeletal muscle troponin T (sTnT) and vitamin D binding protein (DBP) concentrations in 67 the blood, to estimate the degree of muscle degradation in marathon (MR) participants 68 and 100 km adventure race (AR) participants.
This study is concise and concludes with the measurement of the parameters in the blood. It correlates with the literature as well.
Please pay attention to the formatting of the manuscript in terms of spacing.
Reviewer 2 Report
The study reports marathon runners vs adventure race runners blood and serum profile of 9 adult males participating. There are a lot of unaddressed limitations of this micropilot study, that is completely unaddressed. I have the following comments:
1) Why was there no female participants included in the study? This creates an imbalance to any conclusion and even the title can be a bit misleading. I urge the authors to kindly appropriately label their study to avoid confusion.
2) Is there any elevation profile of the marathon route available from fitness app data? If so, kindly plot against the one for AR in Fig 1.
3) CK activity can also depend on the nutrition habit (vegetarian/vegan/non-vegetarian etc) of an individual, besides the type of replenishment they took during the races. Is there any data available for that? Besides, is there any data on other blood parameters, Hb levels, chronic disease history available for the racers?
3) Overall the study is inadequate at the moment to support the experimental design and conclusions. Kindly add any and as much data obtained as possible and also a strength/limitation section in discussion.
Reviewer 3 Report
The ms describes the effect of high endurance activities on skeletal muscle damage. The activity of CK was chosen as a standard measure of muscle damage and two more markers were analyzed, sTNT and DBP. It was shown that immediately after activity the sTNT level increases and DBP level decreases.
The ms is of value and can be published after the following corrections.
Line 74. Is the temperature important? If it is, please provide the temperature of the other race and discuss it. If not, why report it?
Figure 1 is meaningless, please remove or place it in the supplemental material
Section 2.3, please provide the method of serum preparation
Section 2.3, how triplicates were used in the statistical analysis?
Line 108 and below. Striated muscle or skeletal muscle? Please elaborate.
Line 110 and below, please provide the procedure for Ab purification
Line 114, is it “100 μl of 0.05 M carbonate buffer” or something in the buffer?
Line 125, define PBST
Line 130, what means serum concentration?
Line 134, “Creatine kinase was determined” – CK activity perhaps?, please provide the catalog number of the kit
Lines 135-137, cardiac troponins, I, T – provide details on how these proteins were assayed
Line 149 and below, the unit of hematocrit level is %%
Table 2 is hard to read. Please replace it with a bar graph or box (or similar) plot. Why CIs are reported? Please provide raw data in supplemental materials. It is hard to believe that 275.11±71.77 and 236.22±58.60 are significantly different. What is [u.l-1]? Define baseline blood sample.
Lines 157-175, please define F, p, and nu2.
Line 180, CK-MB
Line 189-190, please provide multiday data
Line 208, “a stronger affinity of sTnT to thin filaments” is an ambitious conclusion, since the level of sTnI was not measured in this work.
Line 249, to conclude on the lifetime of the DBP, please provide time-dependent data, not just measurements before and after the activity
General remark (minor), the text suffers from awkward sentences and typos, such as “CK, which is recognized over the years marker of muscle damage is still in use”, or “Sima”, “self-produced”, “the molecules of that protein”
Round 2
Reviewer 1 Report
Thank you for the updates and changes
Reviewer 2 Report
Thank the authors for their explanation and modifications in the manuscript.